# Gradient-free Hamiltonian Monte Carlo with Efficient Kernel Exponential Families

**Heiko Strathmann**[∗] **Dino Sejdinovic**[+] **Samuel Livingstone**[o] **Zoltan Szabo**[∗] **Arthur Gretton**[∗]

[∗]Gatsby Unit
University College London

[+]Department of Statistics
University of Oxford

[o]School of Mathematics
University of Bristol

## Abstract

We propose *Kernel Hamiltonian Monte Carlo (KMC)*, a gradient-free adaptive MCMC algorithm based on Hamiltonian Monte Carlo (HMC). On target densities where classical HMC is not an option due to intractable gradients, KMC adaptively learns the target's gradient structure by fitting an exponential family model in a Reproducing Kernel Hilbert Space. Computational costs are reduced by two novel efficient approximations to this gradient. While being asymptotically exact, KMC mimics HMC in terms of sampling efficiency, and offers substantial mixing improvements over state-of-the-art gradient free samplers. We support our claims with experimental studies on both toy and real-world applications, including Approximate Bayesian Computation and exact-approximate MCMC.

## 1 Introduction

Estimating expectations using Markov Chain Monte Carlo (MCMC) is a fundamental approximate inference technique in Bayesian statistics. MCMC itself can be computationally demanding, and the expected estimation error depends directly on the correlation between successive points in the Markov chain. Therefore, efficiency can be achieved by taking large steps with high probability.

Hamiltonian Monte Carlo [1] is an MCMC algorithm that improves efficiency by exploiting gradient information. It simulates particle movement along the contour lines of a dynamical system constructed from the target density. Projections of these trajectories cover wide parts of the target's support, and the probability of accepting a move along a trajectory is often close to one. Remarkably, this property is mostly invariant to growing dimensionality, and HMC here often is superior to random walk methods, which need to decrease their step size at a much faster rate [1, Sec. 4.4].

Unfortunately, for a large class of problems, gradient information is not available. For example, in Pseudo-Marginal MCMC (PM-MCMC) [2, 3], the posterior does not have an analytic expression, but can only be estimated at any given point, e.g. in Bayesian Gaussian Process classification [4]. A related setting is MCMC for Approximate Bayesian Computation (ABC-MCMC), where the posterior is approximated through repeated simulation from a likelihood model [5, 6]. In both cases, HMC cannot be applied, leaving random walk methods as the only mature alternative. There have been efforts to mimic HMC's behaviour using stochastic gradients from mini-batches in Big Data [7], or stochastic finite differences in ABC [8]. Stochastic gradient based HMC methods, however, often suffer from low acceptance rates or additional bias that is hard to quantify [9].

Random walk methods can be tuned by matching scaling of steps and target. For example, Adaptive Metropolis-Hastings (AMH) [10, 11] is based on learning the global scaling of the target from the history of the Markov chain. Yet, for densities with nonlinear support, this approach does not work very well. Recently, [12] introduced a Kernel Adaptive Metropolis-Hastings (KAMH) algorithm whose proposals are locally aligned to the target. By adaptively learning target covariance in a Reproducing Kernel Hilbert Space (RKHS), KAMH achieves improved sampling efficiency.

In this paper, we extend the idea of using kernel methods to learn efficient proposal distributions [12]. Rather than *locally* smoothing the target density, however, we estimate its gradients *globally*. More precisely, we fit an infinite dimensional exponential family model in an RKHS via score matching [13, 14]. This is a non-parametric method of modelling the log unnormalised target density as an RKHS function, and has been shown to approximate a rich class of density functions arbitrarily well. More importantly, the method has been empirically observed to be relatively robust to increasing dimensionality – in sharp contrast to classical kernel density estimation [15, Sec. 6.5]. Gaussian Processes (GP) were also used in [16] as an emulator of the target density in order to speed up HMC, however, this requires access to the target in closed form, to provide training points for the GP.

We require our adaptive KMC algorithm to be computationally efficient, as it deals with high-dimensional MCMC chains of growing length. We develop two novel approximations to the infinite dimensional exponential family model. The first approximation, *score matching lite*, is based on computing the solution in terms of a lower dimensional, yet growing, subspace in the RKHS. KMC with score matching lite (*KMC lite*) is geometrically ergodic on the same class of targets as standard random walks. The second approximation uses a finite dimensional feature space (*KMC finite*), combined with random Fourier features [17]. KMC finite is an efficient online estimator that allows to use *all* of the Markov chain history, at the cost of decreased efficiency in unexplored regions. A choice between KMC lite and KMC finite ultimately depends on the ability to initialise the sampler within high-density regions of the target; alternatively, the two approaches could be combined.

Experiments show that KMC inherits the efficiency of HMC, and therefore mixes significantly better than state-of-the-art gradient-free adaptive samplers on a number of target densities, including on synthetic examples, and when used in PM-MCMC and ABC-MCMC. All code can be found at `https://github.com/karlnapf/kernel_hmc`

## 2   Background and Previous Work

Let the domain of interest $\mathcal{X}$ be a compact[1] subset of $\mathbb{R}^d$, and denote the unnormalised *target* density on $\mathcal{X}$ by $\pi$. We are interested in constructing a Markov chain $x_1 \rightarrow x_2 \rightarrow \dots$ such that $\lim_{t \rightarrow \infty} x_t \sim \pi$. By running the Markov chain for a long time $T$, we can consistently approximate any expectation w.r.t. $\pi$. Markov chains are constructed using the Metropolis-Hastings algorithm, which at the current state $x_t$ draws a point from a proposal mechanism $x^* \sim Q(\cdot|x_t)$, and sets $x_{t+1} \leftarrow x^*$ with probability $\min(1, [\pi(x^*)Q(x_t|x^*)]/[\pi(x_t)Q(x^*|x_t)])$, and $x_{t+1} \leftarrow x_t$ otherwise. We assume that $\pi$ is intractable,[2] i.e. that we can neither evaluate $\pi(x)$ nor[3] $\nabla \log \pi(x)$ for any $x$, but can only estimate it unbiasedly via $\hat{\pi}(x)$. Replacing $\pi(x)$ with $\hat{\pi}(x)$ results in PM-MCMC [2, 3], which asymptotically remains exact (*exact-approximate inference*).

**(Kernel) Adaptive Metropolis-Hastings**   In the absence of $\nabla \log \pi$, the usual choice of $Q$ is a random walk, i.e. $Q(\cdot|x_t) = \mathcal{N}(\cdot|x_t, \Sigma_t)$. A popular choice of the scaling is $\Sigma_t \propto I$. When the scale of the target density is not uniform across dimensions, or if there are strong correlations, the AMH algorithm [10, 11] improves mixing by adaptively learning global covariance structure of $\pi$ from the history of the Markov chain. For cases where the local scaling does not match the global covariance of $\pi$, i.e. the support of the target is nonlinear, KAMH [12] improves mixing by learning the target covariance in a RKHS. KAMH proposals are Gaussian with a covariance that matches the local covariance of $\pi$ around the current state $x_t$, without requiring access to $\nabla \log \pi$.

**Hamiltonian Monte Carlo**   Hamiltonian Monte Carlo (HMC) uses deterministic, measure-preserving maps to generate efficient Markov transitions [1, 18]. Starting from the negative log target, referred to as the *potential energy* $U(q) = -\log \pi(q)$, we introduce an auxiliary *momentum* variable $p \sim \exp(-K(p))$ with $p \in \mathcal{X}$. The joint distribution of $(p, q)$ is then proportional to $\exp(-H(p, q))$, where $H(p, q) := K(p) + U(q)$ is called the *Hamiltonian*. $H(p, q)$ defines a *Hamiltonian flow*, parametrised by a trajectory length $t \in \mathbb{R}$, which is a map $\phi_t^H : (p, q) \mapsto (p^*, q^*)$ for which $H(p^*, q^*) = H(p, q)$. This allows constructing $\pi$-invariant Markov chains: for a chain at state $q = x_t$, repeatedly (i) re-sample $p' \sim \exp(-K(\cdot))$, and then (ii) apply the Hamiltonian flow

for time $t$, giving $(p^*, q^*) = \phi_t^H(p', q)$. The flow can be generated by the *Hamiltonian operator*

$$\frac{\partial K}{\partial p}\frac{\partial}{\partial q} - \frac{\partial U}{\partial q}\frac{\partial}{\partial p} \qquad (1)$$

In practice, (1) is usually unavailable and we need to resort to approximations. Here, we limit ourselves to the leap-frog integrator; see [1] for details. To correct for discretisation error, a Metropolis acceptance procedure can be applied: starting from $(p', q)$, the end-point of the approximate trajectory is accepted with probability $\min\left[1, \exp\left(-H(p^*, q^*) + H(p', q)\right)\right]$. HMC is often able to propose distant, uncorrelated moves with a high acceptance probability.

**Intractable densities**     In many cases the gradient of $\log \pi(q) = -U(q)$ cannot be written in closed form, leaving random-walk based methods as the state-of-the-art [11, 12]. We aim to overcome random-walk behaviour, so as to obtain significantly more efficient sampling [1].

## 3   Kernel Induced Hamiltonian Dynamics

KMC replaces the potential energy in (1) by a kernel induced surrogate computed from the history of the Markov chain. This surrogate does not require gradients of the log-target density. The surrogate induces a kernel Hamiltonian flow, which can be numerically simulated using standard leap-frog integration. As with the discretisation error in HMC, any deviation of the kernel induced flow from the true flow is corrected via a Metropolis acceptance procedure. This here also contains the estimation noise from $\hat{\pi}$ and re-uses previous values of $\hat{\pi}$, c.f. [3, Table 1]. Consequently, the stationary distribution of the chain remains correct, given that we take care when adapting the surrogate.

**Infinite Dimensional Exponential Families in a RKHS**    We construct a kernel induced potential energy surrogate whose gradients approximate the gradients of the true potential energy $U$ in (1), without accessing $\pi$ or $\nabla \pi$ directly, but only using the history of the Markov chain. To that end, we model the (unnormalised) target density $\pi(x)$ with an infinite dimensional exponential family model [13] of the form

$$\text{const} \times \pi(x) \approx \exp\left(\langle f, k(x, \cdot)\rangle_{\mathcal{H}} - A(f)\right), \qquad (2)$$

which in particular implies $\nabla f \approx -\nabla U = \nabla \log \pi$. Here $\mathcal{H}$ is a RKHS of real valued functions on $\mathcal{X}$. The RKHS has a uniquely associated symmetric, positive definite *kernel* $k : \mathcal{X} \times \mathcal{X} \to \mathbb{R}$, which satisfies $f(x) = \langle f, k(x, \cdot)\rangle_{\mathcal{H}}$ for any $f \in \mathcal{H}$ [19]. The canonical feature map $k(\cdot, x) \in \mathcal{H}$ here takes the role of the *sufficient statistics* while $f \in \mathcal{H}$ are the *natural parameters*, and $A(f) := \log \int_{\mathcal{X}} \exp(\langle f, k(x, \cdot)\rangle_{\mathcal{H}})dx$ is the cumulant generating function. Eq. (2) defines broad class of densities: when universal kernels are used, the family is dense in the space of continuous densities on compact domains, with respect to e.g. Total Variation and KL [13, Section 3]. It is possible to consistently fit an *unnormalised* version of (2) by directly minimising the expected gradient mismatch between the model (2) and the true target density $\pi$ (observed through the Markov chain history). This is achieved by generalising the score matching approach [14] to infinite dimensional parameter spaces. The technique avoids the problem of dealing with the intractable $A(f)$, and reduces the problem to solving a linear system. More importantly, the approach is observed to be relatively robust to increasing dimensions. We return to estimation in Section 4, where we develop two efficient approximations. For now, assume access to an $\hat{f} \in \mathcal{H}$ such that $\nabla f(x) \approx \nabla \log \pi(x)$.

**Kernel Induced Hamiltonian Flow**    We define a kernel induced Hamiltonian operator by replacing $U$ in the potential energy part $\frac{\partial U}{\partial p}\frac{\partial}{\partial q}$ in (1) by our kernel surrogate $U_k = f$. It is clear that, depending on $U_k$, the resulting kernel induced Hamiltonian flow differs from the original one. That said, any bias on the resulting Markov chain, in addition to discretisation error from the leap-frog integrator, is naturally corrected for in the Pseudo-Marginal Metropolis step. We accept an end-point $\phi_t^{H_k}(p', q)$ of a trajectory starting at $(p', q)$ along the *kernel induced* flow with probability

$$\min\left[1, \exp\left(-H\left(\phi_t^{H_k}(p', q)\right) + H(p', q)\right)\right], \qquad (3)$$

where $H\left(\phi_t^{H_k}(p', q)\right)$ corresponds to the *true* Hamiltonian at $\phi_t^{H_k}(p', q)$. Here, in the Pseudo-Marginal context, we replace both terms in the ratio in (3) by unbiased estimates, i.e., we replace

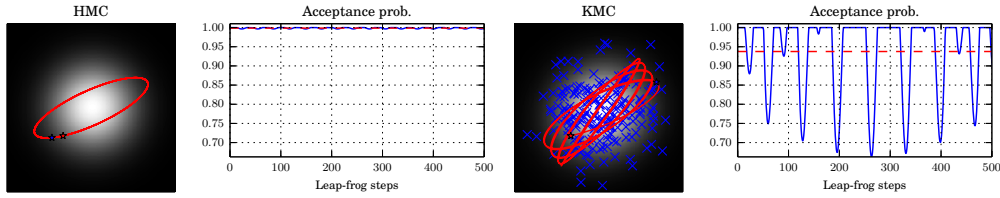

Figure 1: Hamiltonian trajectories on a 2-dimensional standard Gaussian. End points of such trajectories (red stars to blue stars) form the proposal of HMC-like algorithms. **Left:** Plain Hamiltonian trajectories oscillate on a stable orbit, and acceptance probability is close to one. **Right:** Kernel induced trajectories and acceptance probabilities on an estimated energy function.

$\pi(q)$ within $H$ with an unbiased estimator $\hat{\pi}(q)$. Note that this also involves 'recycling' the estimates of $H$ from previous iterations to ensure anyymptotic correctness, c.f. [3, Table 1]. Any deviations of the kernel induced flow from the true flow result in a decreased acceptance probability (3). We therefore need to control the approximation quality of the kernel induced potential energy to maintain high acceptance probability in practice. See Figure 1 for an illustrative example.

## 4    Two Efficient Estimators for Exponential Families in RKHS

We now address estimating the infinite dimensional exponential family model (2) from data. The original estimator in [13] has a large computational cost. This is problematic in the adaptive MCMC context, where the model has to be updated on a regular basis. We propose two efficient approximations, each with its strengths and weaknesses. Both are based on score matching.

### 4.1    Score Matching

Following [14], we model an unnormalised log probability density $\log \pi(x)$ with a parametric model

$$\log \tilde{\pi}_Z(x; f) := \log \tilde{\pi}(x; f) - \log Z(f), \tag{4}$$

where $f$ is a collection of parameters of yet unspecified dimension (c.f. natural parameters of (2)), and $Z(f)$ is an unknown normalising constant. We aim to find $\hat{f}$ from a set of $n$ samples[4] $\mathcal{D} := \{x_i\}_{i=1}^n \sim \pi$ such that $\pi(x) \approx \tilde{\pi}(x; \hat{f}) \times$ const. From [14, Eq. 2], the criterion being optimised is the expected squared distance between gradients of the log density, so-called *score functions*,

$$J(f) = \frac{1}{2} \int_{\mathcal{X}} \pi(x) \left\| \nabla \log \tilde{\pi}(x; f) - \nabla \log \pi(x) \right\|_2^2 dx,$$

where we note that the normalising constants vanish from taking the gradient $\nabla$. As shown in [14, Theorem 1], it is possible to compute an empirical version *without* accessing $\pi(x)$ or $\nabla \log \pi(x)$ other than through observed samples,

$$\hat{J}(f) = \frac{1}{n} \sum_{x \in \mathcal{D}} \sum_{\ell=1}^d \left[ \frac{\partial^2 \log \tilde{\pi}(x; f)}{\partial x_\ell^2} + \frac{1}{2} \left( \frac{\partial \log \tilde{\pi}(x; f)}{\partial x_\ell} \right)^2 \right]. \tag{5}$$

Our approximations of the original model (2) are based on minimising (5) using approximate scores.

### 4.2    Infinite Dimensional Exponential Families Lite

The original estimator of $f$ in (2) takes a dual form in a RKHS sub-space spanned by $nd + 1$ kernel derivatives, [13, Thm. 4]. The update of the proposal at the iteration $t$ of MCMC requires inversion of a $(td + 1) \times (td + 1)$ matrix. This is clearly prohibitive if we are to run even a moderate number of iterations of a Markov chain. Following [12], we take a simple approach to avoid prohibitive computational costs in $t$: we form a proposal using a random sub-sample of fixed size $n$ from the Markov chain history, $\mathbf{z} := \{z_i\}_{i=1}^n \subseteq \{x_i\}_{i=1}^t$. In order to avoid excessive computation when $d$ is large, we replace the full dual solution with a solution in terms of span $(\{k(z_i, \cdot)\}_{i=1}^n)$, which covers the support of the true density by construction, and grows with increasing $n$. That is, we assume that the model (4) takes the 'light' form

$$f(x) = \sum_{i=1}^{n} \alpha_i k(z_i, x), \tag{6}$$

where $\alpha \in \mathbb{R}^n$ are real valued parameters that are obtained by minimising the empirical score matching objective (5). This representation is of a form similar to [20, Section 4.1], the main differences being that the basis functions are chosen randomly, the basis set grows with $n$, and we will require an additional regularising term. The estimator is summarised in the following proposition, which is proved in Appendix A.

**Proposition 1.** *Given a set of samples* $\mathbf{z} = \{z_i\}_{i=1}^{n}$ *and assuming* $f(x) = \sum_{i=1}^{n} \alpha_i k(z_i, x)$ *for the Gaussian kernel of the form* $k(x, y) = \exp\left(-\sigma^{-1}\|x - y\|_2^2\right)$, *and* $\lambda > 0$, *the unique minimiser of the* $\lambda\|f\|_{\mathcal{H}}^2$-*regularised empirical score matching objective* (5) *is given by*

$$\hat{\alpha}_\lambda = -\frac{\sigma}{2}(C + \lambda I)^{-1}b, \tag{7}$$

*where* $b \in \mathbb{R}^n$ *and* $C \in \mathbb{R}^{n \times n}$ *are given by*

$$b = \sum_{\ell=1}^{d} \left( \frac{2}{\sigma}(Ks_\ell + D_{s_\ell}K\mathbf{1} - 2D_{x_\ell}Kx_\ell) - K\mathbf{1} \right) \ and \ C = \sum_{\ell=1}^{d} \left[ D_{x_\ell}K - KD_{x_\ell} \right] \left[ KD_{x_\ell} - D_{x_\ell}K \right],$$

*with entry-wise products* $s_\ell := x_\ell \odot x_\ell$ *and* $D_x := diag(x)$.

The estimator costs $\mathcal{O}(n^3 + dn^2)$ computation (for computing $C, b$, and for inverting $C$) and $\mathcal{O}(n^2)$ storage, for a fixed random chain history sub-sample size $n$. This can be further reduced via low-rank approximations to the kernel matrix and conjugate gradient methods, which are derived in Appendix A.

Gradients of the model are given as $\nabla f(x) = \sum_{i=1}^{n} \alpha_i \nabla k(x, x_i)$, i.e. they simply require to evaluate gradients of the kernel function. Evaluation and storage of $\nabla f(\cdot)$ both cost $\mathcal{O}(dn)$.

### 4.3 Exponential Families in Finite Feature Spaces

Instead of fitting an infinite-dimensional model on a subset of the available data, the second estimator is based on fitting a finite dimensional approximation using *all* available data $\{x_i\}_{i=1}^{t}$, in *primal* form. As we will see, updating the estimator when a new data point arrives can be done online.

Define an $m$-dimensional approximate feature space $\mathcal{H}_m = \mathbb{R}^m$, and denote by $\phi_x \in \mathcal{H}_m$ the embedding of a point $x \in \mathcal{X} = \mathbb{R}^d$ into $\mathcal{H}_m = \mathbb{R}^m$. Assume that the embedding approximates the kernel function as a finite rank expansion $k(x, y) \approx \phi_x^\top \phi_y$. The log unnormalised density of the infinite model (2) can be approximated by assuming the model in (4) takes the form

$$f(x) = \langle \theta, \phi_x \rangle_{\mathcal{H}_m} = \theta^\top \phi_x \tag{8}$$

To fit $\theta \in \mathbb{R}^m$, we again minimise the score matching objective (5), as proved in Appendix B.

**Proposition 2.** *Given a set of samples* $\mathbf{x} = \{x_i\}_{i=1}^{t}$ *and assuming* $f(x) = \theta^\top \phi_x$ *for a finite dimensional feature embedding* $x \mapsto \phi_x \in \mathbb{R}^m$, *and* $\lambda > 0$, *the unique minimiser of the* $\lambda\|\theta\|_2^2$-*regularised empirical score matching objective* (5) *is given by*

$$\hat{\theta}_\lambda := (C + \lambda I)^{-1}b, \tag{9}$$

*where*

$$b := -\frac{1}{n}\sum_{i=1}^{t}\sum_{\ell=1}^{d} \ddot{\phi}_{x_i}^\ell \in \mathbb{R}^m, \qquad C := \frac{1}{n}\sum_{i=1}^{t}\sum_{\ell=1}^{d} \dot{\phi}_{x_i}^\ell \left( \dot{\phi}_{x_i}^\ell \right)^T \in \mathbb{R}^{m \times m},$$

*with* $\dot{\phi}_x^\ell := \frac{\partial}{\partial x_\ell}\phi_x$ *and* $\ddot{\phi}_x^\ell := \frac{\partial^2}{\partial x_\ell^2}\phi_x$.

An example feature embedding based on random Fourier features [17, 21] and a standard Gaussian kernel is $\phi_x = \sqrt{\frac{2}{m}}\left[\cos(\omega_1^T x + u_1), \ldots, \cos(\omega_m^T x + u_m)\right]$, with $\omega_i \sim \mathcal{N}(\omega)$ and $u_i \sim \texttt{Uniform}[0, 2\pi]$. The estimator has a one-off cost of $\mathcal{O}(tdm^2 + m^3)$ computation and $\mathcal{O}(m^2)$ storage. Given that we have computed a solution based on the Markov chain history $\{x_i\}_{i=1}^{t}$, however, it is straightforward to update $C, b$, and the solution $\hat{\theta}_\lambda$ online, after a new point $x_{t+1}$ arrives. This is achieved by storing running averages and performing low-rank updates of matrix inversions, and costs $\mathcal{O}(dm^2)$ computation and $\mathcal{O}(m^2)$ storage, *independent* of $t$. Further details are given in Appendix B.

Gradients of the model are $\nabla f(x) = [\nabla\phi_x]^\top \hat{\theta}$, i.e., they require the evaluation of the gradient of the feature space embedding, costing $\mathcal{O}(md)$ computation and and $\mathcal{O}(m)$ storage.

---

**Algorithm 1 Kernel Hamiltonian Monte Carlo – Pseudo-code**

---

***Input***: Target (possibly noisy estimator) $\hat{\pi}$, adaptation schedule $a_t$, HMC parameters,
     Size of basis $m$ or sub-sample size $n$.
At iteration $t+1$, current state $x_t$, history $\{x_i\}_{i=1}^t$, perform (1-4) with probability $a_t$

**KMC lite:**

1. Update sub-sample $\mathbf{z} \subseteq \{x_i\}_{i=1}^t$
2. Re-compute $C, b$ from Prop. 1
3. Solve $\hat{\alpha}_\lambda = -\frac{\sigma}{2}(C + \lambda I)^{-1} b$
4. $\nabla f(x) \leftarrow \sum_{i=1}^n \alpha_i \nabla k(x, z_i)$

**KMC finite:**

1. Update to $C, b$ from Prop. 2
2. Perform rank-$d$ update to $C^{-1}$
3. Update $\hat{\theta}_\lambda = (C + \lambda I)^{-1} b$
4. $\nabla f(x) \leftarrow [\nabla \phi_x]^\top \hat{\theta}$

5. Propose $(p', x^*)$ with kernel induced Hamiltonian flow, using $\nabla_x U = \nabla_x f$
6. Perform Metropolis step using $\hat{\pi}$: accept $x_{t+1} \leftarrow x^*$ w.p. (3) and reject $x_{t+1} \leftarrow x_t$ otherwise
    If $\hat{\pi}$ is noisy and $x^*$ was accepted, store above $\hat{\pi}(x^*)$ for evaluating (3) in the next iteration

---

## 5 Kernel Hamiltonian Monte Carlo

Constructing a kernel induced Hamiltonian flow as in Section 3 from the gradients of the infinite dimensional exponential family model (2), and approximate estimators (6),(8), we arrive at a gradient free, adaptive MCMC algorithm: *Kernel Hamiltonian Monte Carlo* (Algorithm 1).

**Computational Efficiency, Geometric Ergodicity, and Burn-in** KMC finite using (8) allows for online updates using the *full* Markov chain history, and therefore is a more elegant solution than KMC lite, which has greater computational cost and requires sub-sampling the chain history. Due to the parametric nature of KMC finite, however, the tails of the estimator are not guaranteed to decay. For example, the random Fourier feature embedding described below Proposition 2 contains periodic cosine functions, and therefore oscillates in the tails of (8), resulting in a reduced acceptance probability. As we will demonstrate in the experiments, this problem does not appear when KMC finite is initialised in high-density regions, nor after burn-in. In situations where information about the target density support is unknown, and during burn-in, we suggest to use the lite estimator (7), whose gradients decay outside of the training data. As a result, KMC lite is guaranteed to fall back to a Random Walk Metropolis in unexplored regions, inheriting its convergence properties, and smoothly transitions to HMC-like proposals as the MCMC chain grows. A proof of the proposition below can be found in Appendix C.

**Proposition 3.** *Assume $d = 1$, $\pi(x)$ has log-concave tails, the regularity conditions of [22, Thm 2.2] (implying $\pi$-irreducibility and smallness of compact sets), that MCMC adaptation stops after a fixed time, and a fixed number $L$ of $\epsilon$-leapfrog steps. If $\limsup_{\|x\|_2 \to \infty} \|\nabla f(x)\|_2 = 0$, and $\exists M : \forall x : \|\nabla f(x)\|_2 \leq M$, then KMC lite is geometrically ergodic from $\pi$-almost any starting point.*

**Vanishing adaptation** MCMC algorithms that use the history of the Markov chain for constructing proposals might not be asymptotically correct. We follow [12, Sec. 4.2] and the idea of 'vanishing adaptation' [11], to avoid such biases. Let $\{a_t\}_{i=0}^\infty$ be a schedule of decaying probabilities such that $\lim_{t \to \infty} a_t = 0$ and $\sum_{t=0}^\infty a_t = \infty$. We update the density gradient estimate according to this schedule in Algorithm 1. Intuitively, adaptation becomes less likely as the MCMC chain progresses, but never fully stops, while sharing asymptotic convergence with adaptation that stops at a fixed point [23, Theorem 1]. Note that Proposition 3 is a stronger statement about the *convergence rate*.

**Free Parameters** KMC has two free parameters: the Gaussian kernel bandwidth $\sigma$, and the regularisation parameter $\lambda$. As KMC's performance depends on the quality of the approximate infinite dimensional exponential family model in (6) or (8), a principled approach is to use the score matching objective function in (5) to choose $\sigma, \lambda$ pairs via cross-validation (using e.g. 'hot-started' blackbox optimisation). Earlier adaptive kernel-based MCMC methods [12] did not address parameter choice.

## 6 Experiments

We start by quantifying performance of KMC finite on synthetic targets. We emphasise that these results can be reproduced with the lite version.

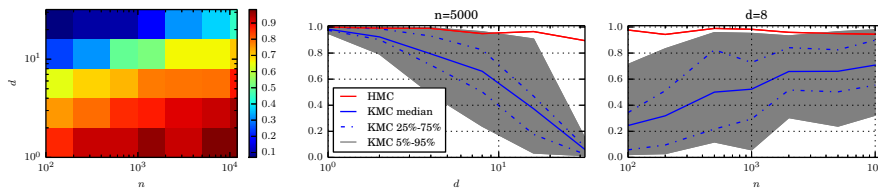

Figure 2: Hypothetical acceptance probability of KMC finite on a challening target in growing dimensions. **Left:** As a function of $n = m$ (x-axis) and $d$ (y-axis). **Middle/right:** Slices through left plot with error bars for fixed $n = m$ and as a function of $d$ (left), and for fixed $d$ as a function of $n = m$ (right).

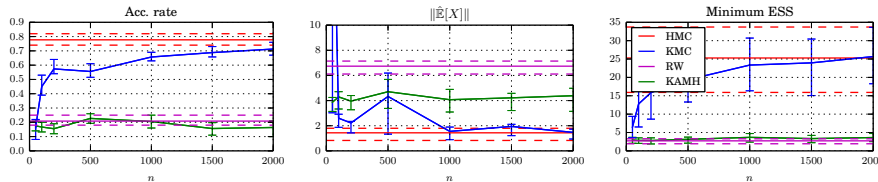

Figure 3: Results for the 8-dimensional synthetic Banana. As the amout of observed data increases, KMC performance approaches HMC – outperforming KAMH and RW. 80% error bars over 30 runs.

**KMC Finite: Stability of Trajectories in High Dimensions** In order to quantify efficiency in growing dimensions, we study hypothetical acceptance rates along trajectories on the kernel induced Hamiltonian flow (no MCMC yet) on a challenging Gaussian target: We sample the diagonal entries of the covariance matrix from a `Gamma(1,1)` distribution and rotate with a uniformly sampled random orthogonal matrix. The resulting target is challenging to estimate due to its 'non-singular smoothness', i.e., substantially differing length-scales across its principal components. As a single Gaussian kernel is not able to effeciently represent such scaling families, we use a rational quadratic kernel for the gradient estimation, whose random features are straightforward to compute. Figure 2 shows the average acceptance over 100 independent trials as a function of the number of (ground truth) samples and basis functions, which are set to be equal $n = m$, and of dimension $d$. In low to moderate dimensions, gradients of the finite estimator lead to acceptance rates comparable to plain HMC. On targets with more 'regular' smoothness, the estimator performs well in up to $d \approx 100$, with less variance. See Appendix D.1 for details.

**KMC Finite: HMC-like Mixing on a Synthetic Example** We next show that KMC's performance approaches that of HMC as it sees more data. We compare KMC, HMC, an isotropic random walk (RW), and KAMH on the 8-dimensional nonlinear banana-shaped target; see Appendix D.2. We here only quantify mixing *after* a sufficient burn-in (burn-in speed is included in next example). We quantify performance on estimating the target's mean, which is exactly **0**. We tuned the scaling of KAMH and RW to achieve 23% acceptance. We set HMC parameters to achieve 80% acceptance and then used the same parameters for KMC. We ran all samplers for 2000+200 iterations from a random start point, discarded the burn-in and computed acceptance rates, the norm of the empirical mean $\|\hat{\mathbb{E}}[x]\|$, and the minimum effective sample size (ESS) across dimensions. For KAMH and KMC, we repeated the experiment for an increasing number of burn-in samples and basis functions $m = n$. Figure 3 shows the results as a function of $m = n$. KMC clearly outperforms RW and KAMH, and eventually achieves performance close to HMC as $n = m$ grows.

**KMC Lite: Pseudo-Marginal MCMC for GP Classification on Real World Data** We next apply KMC to sample from the marginal posterior over hyper-parameters of a Gaussian Process Classification (GPC) model on the UCI Glass dataset [24]. Classical HMC cannot be used for this problem, due to the intractability of the marginal data likelihood. Our experimental protocol mostly follows [12, Section 5.1], see Appendix D.3, but uses only 6000 MCMC iterations *without* discarding a burn-in, i.e., we study how fast KMC initially explores the target. We compare convergence in terms of all mixed moments of order up to 3 to a set of benchmark samples (MMD [25], lower is better). KMC randomly uses between 1 and 10 leapfrog steps of a size chosen uniformly in $[0.01, 0.1]$,

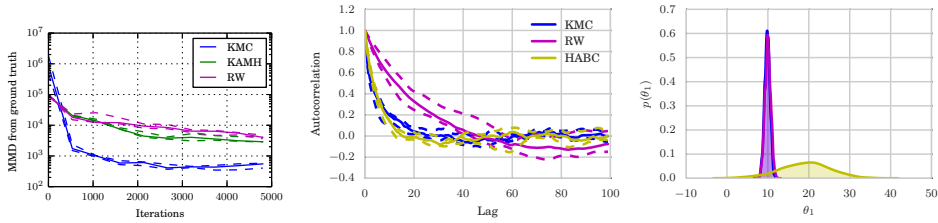

Figure 4: **Left:** Results for 9-dimensional marginal posterior over length scales of a GPC model applied to the UCI Glass dataset. The plots shows convergence (no burn-in discarded) of all mixed moments up to order 3 (lower MMD is better). **Middle/right:** ABC-MCMC auto-correlation and marginal $\theta_1$ posterior for a 10-dimensional skew normal likelihood. While KMC mixes as well as HABC, it does not suffer from any bias (overlaps with RW, while HABC is significantly different) and requires fewer simulations per proposal.

a standard Gaussian momentum, and a kernel tuned by cross-validation, see Appendix D.3. We did not extensively tune the HMC parameters of KMC as the described settings were sufficient. Both KMC and KAMH used 1000 samples from the chain history. Figure 4 (left) shows that KMC's burn-in contains a short 'exploration phase' where produced estimates are bad, due to it falling back to a random walk in unexplored regions, c.f. Proposition 3. From around 500 iterations, however, KMC clearly outperforms both RW and the earlier state-of-the-art KAMH. These results are backed by the minimum ESS (not plotted), which is around 415 for KMC and is around 35 and 25 for KAMH and RW, respectively. Note that all samplers effectively stop improving from 3000 iterations – indicating a burn-in bias. All samplers took 1h time, with most time spent estimating the marginal likelihood.

**KMC Lite: Reduced Simulations and no Additional Bias in ABC**   We now apply KMC in the context of Approximate Bayesian Computation (ABC), which often is employed when the data likelihood is intractable but can be obtained by simulation, see e.g. [6]. ABC-MCMC [5] targets an approximate posterior by constructing an unbiased Monte Carlo estimator of the approximate likelihood. As each such evaluation requires expensive simulations from the likelihood, the goal of all ABC methods is to reduce the number of such simulations. Accordingly, Hamiltonian ABC was recently proposed [8], combining the synthetic likelihood approach [26] with gradients based on stochastic finite differences. We remark that this requires to simulate from the likelihood in *every* leapfrog step, and that the additional bias from the Gaussian likelihood approximation can be problematic. In contrast, KMC does not require simulations to construct a proposal, but rather 'invests' simulations into an accept/reject step (3) that ensures convergence to the *original* ABC target. Figure 4 (right) compares performance of RW, HABC (sticky random numbers and SPAS, [8, Sec. 4.3, 4.4]), and KMC on a 10-dimensional skew-normal distribution $p(y|\theta) = 2\mathcal{N}\left(\theta, I\right)\Phi\left(\langle\alpha, y\rangle\right)$ with $\theta = \alpha = \mathbf{1} \cdot 10$. KMC mixes as well as HABC, but HABC suffers from a severe bias. KMC also reduces the number of simulations per proposal by a factor $2L = 100$. See Appendix D.4 for details.

## 7   Discussion

We have introduced KMC, a kernel-based gradient free adaptive MCMC algorithm that mimics HMC's behaviour by estimating target gradients in an RKHS. In experiments, KMC outperforms random walk based sampling methods in up to $d = 50$ dimensions, including the recent kernel-based KAMH [12]. KMC is particularly useful when gradients of the target density are unavailable, as in PM-MCMC or ABC-MCMC, where classical HMC cannot be used. We have proposed two efficient empirical estimators for the target gradients, each with different strengths and weaknesses, and have given experimental evidence for the robustness of both.

Future work includes establishing theoretical consistency and uniform convergence rates for the empirical estimators, for example via using recent analysis of random Fourier Features with tight bounds [21], and a thorough experimental study in the ABC-MCMC context where we see a lot of potential for KMC. It might also be possible to use KMC as a precomputing strategy to speed up classical HMC as in [27]. For code, see `https://github.com/karlnapf/kernel_hmc`

## Footnotes

[1] The compactness restriction is imposed to satisfy the assumptions in [13].

[2] $\pi$ is analytically intractable, as opposed to computationally expensive in the Big Data context.

[3] Throughout the paper $\nabla$ denotes the gradient operator w.r.t. to $x$.

[4]We assume a fixed sample set here but will use both the full chain history $\{x_i\}_{i=1}^t$ or a sub-sample later.

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
