[Supplementary Material · Gradient-free Hamiltonian Monte Carlowith Efficient Kernel Exponential Families Appendix.pdf]

# Appendix

The Appendix contains proofs for Propositions 1 and 2, as well as additional computational details for both KMC lite in Section A and KMC finite in Section B. Section C covers the proof of geometric ergodicity of KMC lite from Proposition 3. Section D describes further experimental details.

## A Lite Estimator

**Proof of Proposition 1**

The proof below extends the model in [20, Section 4.1]. We assume that the model log-density (4) takes the form in Proposition 1, then directly implement score functions (5), from which we derive an empirical score matching objective as a system of linear equations.

*Proof.* As assumed the log unnormalised density takes the form

$$f(x) = \sum_{i=1}^{n} \alpha_i k(x_i, x)$$

where $k : \mathbb{R}^d \times \mathbb{R}^d \to \mathbb{R}$ is the Gaussian kernel in the form

$$k(x_i, x) = \exp\left(-\frac{1}{\sigma}\|x_i - x\|^2\right) = \exp\left(-\frac{1}{\sigma}\sum_{\ell=1}^{d}(x_{i\ell} - x_\ell)^2\right).$$

The score functions for (5) are then given by

$$\psi_\ell(x; \alpha) := \frac{\partial \log \tilde{\pi}(x; f)}{\partial x_\ell} = \frac{2}{\sigma}\sum_{i=1}^{n}\alpha_i(x_{i\ell} - x_\ell)\exp\left(-\frac{\|x_i - x\|^2}{\sigma}\right),$$

and

$$
\begin{aligned}
\partial_\ell \psi_\ell(x; \alpha) :=& \frac{\partial^2 \log \tilde{\pi}(x; f)}{\partial^2 x_\ell} \\
=& -\frac{2}{\sigma}\sum_{i=1}^{n}\alpha_i \exp\left(-\frac{\|x_i - x\|^2}{\sigma}\right) + \left(\frac{2}{\sigma}\right)^2 \sum_{i=1}^{n}\alpha_i(x_{i\ell} - x_\ell)^2 \exp\left(-\frac{\|x_i - x\|^2}{\sigma}\right) \\
=& \frac{2}{\sigma}\sum_{i=1}^{n}\alpha_i \exp\left(-\frac{\|x_i - x\|^2}{\sigma}\right)\left[-1 + \frac{2}{\sigma}(x_{i\ell} - x_\ell)^2\right].
\end{aligned}
$$

Substituting this into (5) yields

$$
\begin{aligned}
J(\alpha) =& \frac{1}{n}\sum_{i=1}^{n}\sum_{\ell=1}^{d}\left[\partial_\ell \psi_\ell(x_i; \alpha) + \frac{1}{2}\psi_\ell(x_i; \alpha)^2\right] \\
=& \frac{2}{n\sigma}\sum_{\ell=1}^{d}\sum_{i=1}^{n}\sum_{j=1}^{n}\alpha_i \exp\left(-\frac{\|x_i - x_j\|^2}{\sigma}\right)\left[-1 + \frac{2}{\sigma}(x_{i\ell} - x_{j\ell})^2\right] \\
&+ \frac{2}{n\sigma^2}\sum_{\ell=1}^{d}\sum_{i=1}^{n}\left[\sum_{j=1}^{n}\alpha_j(x_{j\ell} - x_{i\ell})\exp\left(-\frac{\|x_i - x_j\|^2}{\sigma}\right)\right]^2.
\end{aligned}
$$

We now rewrite $J(\alpha)$ in matrix form. The expression for the term $J(\alpha)$ being optimised is the sum of two terms.

**First Term**:

$$\sum_{\ell=1}^{d}\sum_{i=1}^{n}\sum_{j=1}^{n}\alpha_i \exp\left(-\frac{\|x_i - x_j\|^2}{\sigma}\right)\left[-1 + \frac{2}{\sigma}(x_{i\ell} - x_{j\ell})^2\right]$$

We only need to compute

$$\sum_{i=1}^{n}\sum_{j=1}^{n}\alpha_i \exp\left(-\frac{\|x_i - x_j\|^2}{\sigma}\right)(x_{i\ell} - x_{j\ell})^2$$

$$=\sum_{i=1}^{n}\sum_{j=1}^{n}\alpha_i \exp\left(-\frac{\|x_i - x_j\|^2}{\sigma}\right)\left(x_{i\ell}^2 + x_{j\ell}^2 - 2x_{i\ell}x_{j\ell}\right).$$

Define

$$x_\ell := \begin{bmatrix} x_{1\ell} & \dots & x_{m\ell} \end{bmatrix}^\top.$$

The final term may be computed with the right ordering of operations,

$$-2(\alpha \odot x_\ell)^\top K x_\ell,$$

where $\alpha \odot x_\ell$ is the entry-wise product. The remaining terms are sums with constant row or column terms. Define $s_\ell := x_\ell \odot x_\ell$ with components $s_{i\ell} = x_{i\ell}^2$. Then

$$\sum_{i=1}^{n}\sum_{j=1}^{n}\alpha_i k_{ij} s_{j\ell} = \alpha^\top K s_\ell.$$

Likewise

$$\sum_{i=1}^{n}\sum_{j=1}^{n}\alpha_i x_{i\ell}^2 k_{ij} = (\alpha \odot s_\ell)^\top K \mathbf{1}.$$

**Second Term**: Considering only the $\ell$-th dimension, this is

$$\sum_{i=1}^{n}\left[\sum_{j=1}^{n}\alpha_j(x_{j\ell} - x_{i\ell})\exp\left(-\frac{\|x_i - x_j\|^2}{\sigma}\right)\right]^2.$$

In matrix notation, the inner sum is a column vector,

$$K(\alpha \odot x_\ell) - (K\alpha) \odot x_\ell.$$

We take the entry-wise square and sum the resulting vector. Denote by $D_x := \mathrm{diag}(x)$, then the following two relations hold

$$K(\alpha \odot x) = K D_x \alpha,$$
$$(K\alpha) \odot x = D_x K \alpha.$$

This means that $J(\alpha)$ as defined previously,

$$J(\alpha) = \frac{2}{n\sigma}\sum_{\ell=1}^{d}\left[\frac{2}{\sigma}\left[\alpha^T K s_\ell + (\alpha \odot s_\ell)^\top K \mathbf{1} - 2(\alpha \odot x_\ell)^\top K x_\ell\right] - \alpha^T K \mathbf{1}\right]$$

$$+ \frac{2}{n\sigma^2}\sum_{\ell=1}^{d}\left[(\alpha \odot x_\ell)^\top K - x_\ell^\top \odot (\alpha^\top K)\right]\left[K(\alpha \odot x_\ell) - (K\alpha) \odot x_\ell\right],$$

can be rewritten as

$$J(\alpha) = \frac{2}{n\sigma}\alpha^T \sum_{\ell=1}^{d}\left[\frac{2}{\sigma}(K s_\ell + D_{s_\ell} K \mathbf{1} - 2 D_{x_\ell} K x_\ell) - K\mathbf{1}\right]$$

$$+ \frac{2}{n\sigma^2}\alpha^T\left(\sum_{\ell=1}^{d}[D_{x_\ell} K - K D_{x_\ell}][K D_{x_\ell} - D_{x_\ell} K]\right)\alpha$$

$$= \frac{2}{n\sigma}\alpha^T b + \frac{2}{n\sigma^2}\alpha^\top C \alpha,$$

where

$$b = \sum_{\ell=1}^{d} \left( \frac{2}{\sigma} (Ks_\ell + D_{s_\ell} K\mathbf{1} - 2D_{x_\ell} Kx_\ell) - K\mathbf{1} \right) \in \mathbb{R}^n,$$

$$C = \sum_{\ell=1}^{d} [D_{x_\ell} K - K D_{x_\ell}] [K D_{x_\ell} - D_{x_\ell} K] \in \mathbb{R}^{n \times n}.$$

Assuming $C$ is invertible, this is minimised by

$$\hat{\alpha} = -\frac{\sigma}{2} C^{-1} b.$$

$\square$

As in [13], we add a term $\lambda \|f\|_{\mathcal{H}}^2$ for $\lambda \in \mathbb{R}^+$, in order to control the norm of the natural parameters in the RKHS $\|f\|_{\mathcal{H}}^2$. This results in the regularised and numerically more stable solution $\hat{\alpha}_\lambda := (C + \lambda I)^{-1} b$.

### Reduced Computational Costs via Low-rank Approximations and Conjugate Gradient

Solving the linear system in (7) requires $\mathcal{O}(n^3)$ computation and $\mathcal{O}(n^2)$ storage for a fixed random sub-sample of the chain history $\mathbf{z}$. In order to allow for large $n$, and to exploit potential manifold structure in the RKHS, we apply a low-rank approximation to the kernel matrix via incomplete Cholesky [28, Alg. 5.12], that is a standard way to achieve linear computational costs for kernel methods. We rewrite the kernel matrix

$$K \approx LL^\top,$$

where $L \in \mathbb{R}^{n \times \ell}$ is obtained via dual partial Gram–Schmidt orthonormalisation and costs both $\mathcal{O}(n\ell)$ computation and storage. Usually $\ell \ll n$, and $\ell$ can be chosen via an accuracy cut-off parameter on the kernel spectrum in the same fashion as for other low-rank approximations, such as PCA[5]. Given such a representation of $K$, we can rewrite any matrix-vector product as

$$Kb \approx (LL^\top)b = L(L^\top b),$$

where each left multiplication of $L$ costs $\mathcal{O}(n\ell)$ and we never need to store $LL^\top$. This idea can be used to achieve costs of $\mathcal{O}(n\ell)$ when computing $b$, and left-multiplying $C$. Combining the technique with conjugate gradient (CG) allows to solve (7) with a maximum of $n$ such matrix-vector products, yielding a total computational cost of $\mathcal{O}(n^2\ell)$. In practice, we can monitor residuals and stop CG after a fixed number of iterations $\tau \ll n$, where $\tau$ depends on the decay of the spectrum of $K$. We arrive at a total cost of $\mathcal{O}(n\ell\tau)$ computation and $\mathcal{O}(n\ell)$ storage. CG also has the advantage of allowing for 'hot starts', i.e. initialising the linear solver at a previous solution. Further details can be found in our implementation.

## B  Finite Feature Space Estimator

### Proof of Proposition 2

We assume the model log-density (4) takes the primal form in a finite dimensional feature space as in Proposition 2, then again directly implement score functions in (5) and minimise it via a linear solve.

*Proof.* As assumed the log unnormalised density takes the form

$$f(x) = \langle \theta, \phi_x \rangle_{\mathcal{H}_m} = \theta^\top \phi_x,$$

where $x \in \mathbb{R}^d$ is embedded into a finite dimensional feature space $\mathcal{H}_m = \mathbb{R}^m$ as $x \mapsto \phi_x$. The score functions in (5) then can be written as the simple linear form

$$\psi_\ell(\xi; \theta) := \frac{\partial \log \tilde{\pi}(x; \theta)}{\partial x_\ell} = \theta^\top \dot{\phi}_x^\ell \quad \text{and} \quad \partial_\ell \psi_\ell(\xi; \theta) := \frac{\partial^2 \log \tilde{\pi}(x; \theta)}{\partial x_\ell^2} = \theta^\top \ddot{\phi}_x^\ell, \qquad (10)$$

where we defined the $m$-dimensional feature vector derivatives $\dot{\phi}_x^\ell := \frac{\partial}{\partial x_\ell} \phi_x$ and $\ddot{\phi}_x^\ell := \frac{\partial^2}{\partial x_\ell^2} \phi_x$. Plugging those into the empirical score matching objective in (5), we arrive at

$$\begin{aligned} J(\theta) &= \frac{1}{n} \sum_{i=1}^n \sum_{\ell=1}^d \left[ \partial_\ell \psi_\ell(x_i; \theta) + \frac{1}{2} \psi_\ell^2(x_i; \theta) \right] \\ &= \frac{1}{n} \sum_{i=1}^n \sum_{\ell=1}^d \left[ \theta^T \ddot{\phi}_{x_i}^\ell + \frac{1}{2} \theta^T \left( \dot{\phi}_{x_i}^\ell \left( \dot{\phi}_{x_i}^\ell \right)^T \right) \theta \right] \\ &= \frac{1}{2} \theta^T C \theta - \theta^T b \end{aligned} \qquad (11)$$

where

$$b := -\frac{1}{n} \sum_{i=1}^n \sum_{\ell=1}^d \ddot{\phi}_{x_i}^\ell \in \mathbb{R}^m \quad \text{and} \quad C := \frac{1}{n} \sum_{i=1}^n \sum_{\ell=1}^d \left( \dot{\phi}_{x_i}^\ell \left( \dot{\phi}_{x_i}^\ell \right)^\top \right) \in \mathbb{R}^{m \times m}. \qquad (12)$$

Assuming that $C$ is invertible (trivial for $n \geq m$), the objective is uniquely minimised by differentiating (11) wrt. $\theta$, setting to zero, and solving for $\theta$. This gives

$$\hat{\theta} := C^{-1} b. \qquad (13)$$

$\square$

Again, similar to [13], we add a term $\lambda/2 \|\theta\|_2^2$ for $\lambda \in \mathbb{R}^+$ to (11), in order to control the norm of the natural parameters $\theta \in \mathcal{H}^m$. This results in the regularised and numerically more stable solution $\hat{\theta}_\lambda := (C + \lambda I)^{-1} b$.

Next, we give an example for the approximate feature space $\mathcal{H}_m$. Note that the above approach can be combined with *any* set of finite dimensional approximate feature mappings $\phi_x$.

**Example: Random Fourier Features for the Gaussian Kernel**

We now combine the finite dimensional approximate infinite dimensional exponential family model with the "random kitchen sink" [17]. Assume a translation invariant kernel $k(x, y) = \tilde{k}(x - y)$. Bochner's theorem gives the representation

$$k(x, y) = \tilde{k}(x - y) = \int_{\mathbb{R}^d} \exp \left( i \omega^\top (x - y) \right) \mathrm{d}\Gamma(\omega),$$

where $\Gamma(\omega)$ is the Fourier transform of the kernel. An approximate feature mapping for such kernels can be obtained via dropping imaginary terms and approximating the integral with Monte Carlo integration. This gives

$$\phi_x = \sqrt{\frac{2}{m}} \left[ \cos(\omega_1^\top x + u_1), \ldots, \cos(\omega_m^\top x + u_m) \right],$$

with fixed random basis vector realisations that depend on the kernel via $\Gamma(\omega)$,

$$\omega_i \sim \Gamma(\omega),$$

and fixed random offset realisations

$$u_i \sim \texttt{Uniform}[0, 2\pi],$$

for $i = 1 \ldots m$. It is easy to see that this approximation is consistent for $m \to \infty$, i.e.

$$\mathbb{E}_{\omega, b} \left[ \phi_x^T \phi_y \right] = k(x, y).$$

See [17] for details and a uniform convergence bound and [21] for a more detailed analysis with tighter bounds. Note that it is possible to achieve logarithmic computational costs in $d$ exploiting properties of Hadamard matrices [29].

The feature map derivatives (10) are given by

$$
\begin{aligned}
\dot{\phi}_\xi^\ell &= \sqrt{\frac{2}{m}} \frac{\partial}{\partial \xi_\ell} \left[ \cos(\omega_1^T \xi + u_1), \dots, \cos(\omega_m^T \xi + u_m) \right] \\
&= -\sqrt{\frac{2}{m}} \left[ \sin(\omega_1^T \xi + u_1)\omega_{1\ell}, \dots, \sin(\omega_m^T \xi + u_m)\omega_{m\ell} \right] \\
&= -\sqrt{\frac{2}{m}} \left[ \sin(\omega_1^T \xi + u_1), \dots, \sin(\omega_m^T \xi + u_m) \right] \odot \left[ \omega_{1\ell}, \dots, \omega_{m\ell} \right],
\end{aligned}
$$

where $\omega_{j\ell}$ is the $\ell$-th component of $\omega_j$, and

$$
\begin{aligned}
\ddot{\phi}_\xi^\ell &:= -\sqrt{\frac{2}{m}} \frac{\partial}{\partial \xi_\ell} \left[ \sin(\omega_1^T \xi + u_1), \dots, \sin(\omega_m^T \xi + u_m) \right] \odot \left[ \omega_{1\ell}, \dots, \omega_{m\ell} \right] \\
&= -\sqrt{\frac{2}{m}} \left[ \cos(\omega_1^T \xi + u_1), \dots, \cos(\omega_m^T \xi + u_m) \right] \odot \left[ \omega_{1\ell}^2, \dots, \omega_{m\ell}^2 \right] \\
&= -\phi_\xi \odot \left[ \omega_{1\ell}^2, \dots, \omega_{m\ell}^2 \right],
\end{aligned}
$$

where $\odot$ is the element-wise product. Consequently the gradient is given by

$$
\nabla_\xi \phi_\xi = \begin{bmatrix} \dot{\phi}_\xi^1 \\ \vdots \\ \dot{\phi}_\xi^d \end{bmatrix} \in \mathbb{R}^{d \times m}.
$$

As an example, the translation invariant Gaussian kernel and its Fourier transform are

$$
k(x, y) = \exp\left( -\frac{\|x - y\|_2^2}{2\sigma^2} \right) \quad \text{and} \quad \Gamma(\omega) = \mathcal{N}\left( \omega \Big| \mathbf{0}, \frac{2}{\sigma^2} I_m \right).
$$

**Constant Cost Updates**

A convenient property of the finite feature space approximation is that its primal representation of the solution allows to update (12) in an online fashion. When combined with MCMC, each new point $x_{t+1}$ of the Markov chain history only adds a term of the form $-\sum_{\ell=1}^d \ddot{\phi}_{x_{t+1}}^\ell \in \mathbb{R}^m$ and $\sum_{\ell=1}^d \dot{\phi}_{x_{t+1}}^\ell (\dot{\phi}_{x_{t+1}}^\ell)^\top \in \mathbb{R}^{m \times m}$ to the moving averages of $b$ and $C$ respectively. Consequently, at iteration $t$, rather than fully re-computing (13) at the cost of $\mathcal{O}(tdm^2 + m^3)$ for every new point, we can use rank-$d$ updates to construct the minimiser of (11) from the solution of the previous iteration. Assume we have computed the sum of all moving average terms,

$$
\bar{C}_t^{-1} := \left( \sum_{i=1}^t \sum_{\ell=1}^d \left( \dot{\phi}_{x_i}^\ell \left( \dot{\phi}_{x_i}^\ell \right)^\top \right) \right)^{-1}
$$

from feature vectors derivatives $\ddot{\phi}_{x_i}^\ell \in \mathbb{R}^m$ of some set of points $\{x_i\}_{i=1}^t$, and subsequently receive receive a new point $x_{t+1}$. We can then write the inverse of the new sum as

$$
\bar{C}_{t+1}^{-1} := \left( \bar{C}_t + \sum_{\ell=1}^d \left( \dot{\phi}_{x_{t+1}}^\ell \left( \dot{\phi}_{x_{t+1}}^\ell \right)^\top \right) \right)^{-1}.
$$

This is the inverse of the rank-$d$ perturbed previous matrix $\bar{C}_t$. We can therefore construct this inverse using $d$ successive applications of the Sherman-Morrison-Woodbury formula for rank-one updates, each using $\mathcal{O}(m^2)$ computation. Since $\bar{C}_t$ is positive definite[6], we can represent its inverse as a numerically much more stable Cholesky factorisation $\bar{C}_t = \bar{L}_t \bar{L}_t^\top$. It is also possible to perform

cheap rank-$d$ updates of such Cholesky factors[7]. Denote by $\bar{b}_t$ the sum of the moving average $b$. We solve (13) as

$$\hat{\theta} = C^{-1}b = \left(\frac{1}{t}\bar{C}_t\right)^{-1}\left(\frac{1}{t}\bar{b}_t\right) = \bar{C}_t^{-1}\bar{b}_t = \bar{L}_t^{-\top}\bar{L}_t^{-1}\bar{b}_t,$$

using cheap triangular back-substitution from $\bar{L}_t$, and never storing $\bar{C}_t^{-1}$ or $\bar{L}_t^{-1}$ explicitly.

Using such updates, the computational costs for updating the approximate infinite dimensional exponential family model in *every* iteration of the Markov chain are $\mathcal{O}(dm^2)$, which *constant in t*. We can therefore use *all* points in the history for constructing a proposal. See our implementation for further details.

**Algorithmic Description:**

1. Update sums

$$\bar{b}_{t+1} = \bar{b}_t - \sum_{\ell=1}^{d} \ddot{\phi}_{x_{t+1}}^\ell \quad \text{and} \quad \bar{C}_{t+1} = \bar{C}_t + \frac{1}{2}\sum_{\ell=1}^{d} \dot{\phi}_{x_{t+1}}^\ell (\dot{\phi}_{x_{t+1}}^\ell)^\top.$$

2. Perform rank-$d$ update to obtain updated Cholesky factorisation $\bar{L}_{t+1}\bar{L}_{t+1}^T = \bar{C}_{t+1}$.
3. Update approximate infinite dimensional exponential family parameters

$$\hat{\theta} = \bar{L}_{t+1}^{-\top}\bar{L}_{t+1}^{-1}\bar{b}_{t+1}.$$

## C   Ergodicity of KMC lite

**Notation**   Denote by $\alpha(x_t, x^*(p'))$ the probability of accepting a $(p', x^*)$ proposal at state $x_t$. Let $a \wedge b = \min(a, b)$. Define $c(x^{(0)}) := L\epsilon^2 \nabla \log \pi(x^{(0)})/2 + \epsilon^2 \sum_{i=1}^{L-1}(L-i)\nabla \log \pi(x^{(i\epsilon)})$ and $d(x^{(0)}) := \epsilon(\nabla f(x^{(0)}) + \nabla f(x^{(L\epsilon)}))/2 + \epsilon \sum_{i=1}^{L-1} \nabla f(x^{(i\epsilon)})$, where $x^{(i\epsilon)}$ is the $i$-th point of the leapfrog integration from $x = x^{(0)}$.

**Proof of Proposition 3**

*Proof.* We assumed $\pi(x)$ is log-concave in the tails, meaning $\exists x_U > 0$ s.t. for $x^* > x_t > x_U$, we have $\pi(x^*)/\pi(x_t) \leq e^{-\alpha_1(\|x^*\|_2 - \|x_t\|_2)}$ and for $x_t > x^* > x_U$, we have $\pi(x^*)/\pi(x_t) \geq e^{-\alpha_1(\|x^*\|_2 - \|x_t\|_2)}$, and a similar condition holds in the negative tail. Furthermore, we assumed fixed HMC parameters: $L$ leapfrog steps of size $\epsilon$, and wlog the identity mass matrix $I$. Following [22, 30], it is sufficient to show

$$\limsup_{\|x_t\|_2 \to \infty} \int \left[e^{s(\|x^*(p')\|_2 - \|x_t\|_2)} - 1\right]\alpha(x_t, x^*(p'))\mu(dp') < 0,$$

for some $s > 0$, where $\mu(\cdot)$ is a standard Gaussian measure. Denoting the integral $I_{-\infty}^\infty$, we split it into

$$I_{-\infty}^{-x_t^\delta} + I_{-x_t^\delta}^{x_t^\delta} + I_{x_t^\delta}^\infty,$$

for some $\delta \in (0, 1)$. We show that the first and third terms decay to zero whilst the second remains strictly negative as $x_t \to \infty$ (a similar argument holds as $x_t \to -\infty$). We detail the case $\nabla f(x) \uparrow 0$ as $x \to \infty$ here, the other is analogous. Taking $I_{-x_t^\delta}^{x_t^\delta}$, we can choose an $x_t$ large enough that $x_t - C - L\epsilon x_t^\delta > x_U$, $-\gamma_1 < c(x_t - x_t^\delta) < 0$ and $-\gamma_2 < d(x_t - x_t^\delta) < 0$. So for $p' \in (0, x_t^\delta)$ we have

$$L\epsilon p' > x^* - x_t > L\epsilon p' - \gamma_1 \implies e^{-\alpha_1(-\gamma_1 + L\epsilon p')} \geq e^{-\alpha_1(x^* - x_t)} \geq \pi(x^*)/\pi(x_t),$$

where the last inequality comes from the log-concave tails assumption. For $p' \in (\gamma_2^2/2, x_t^\delta)$

$$\alpha(x_t, x^*) \leq 1 \wedge \frac{\pi(x^*)}{\pi(x_t)}\exp\left(p'\gamma_2/2 - \gamma_2^2/2\right) \leq 1 \wedge \exp\left(-\alpha_2 p' + \alpha_1\gamma_1 - \gamma_2^2/2\right),$$

Figure 5: Acceptance probability of kernel induced Hamiltonian flow for a standard Gasussian in high dimensions for an isotropic Gaussian. **Left:** As a function of $n = m$ (x-axis) and $d$ (y-axis). **Middle:** Slices through left plot with error bars for a fixed $n = m$ and as a function in $d$ (left), and for a fixed $d$ as a function of $n = m$ (right).

where $x_t$ is large enough that $\alpha_2 = \alpha_1 L\epsilon - \gamma_2/2 > 0$. Similarly for $p' \in (\gamma_1/L\epsilon, x_t^\delta)$

$$e^{sL\epsilon p'} - 1 \geq e^{s(x^* - x_t)} - 1 \geq e^{s(L\epsilon p' - \gamma_1)} - 1 > 0.$$

Because $\gamma_1$ and $\gamma_2$ can be chosen to be arbitrarily small, then for large enough $x_t$ we will have

$$0 < I_0^{x_t^\delta} \leq \int_{\gamma_1/L\epsilon}^{x_t^\delta} [e^{sL\epsilon p'} - 1] \exp\left(-\alpha_2 p' + \alpha_1 \gamma_1 - \gamma_2^2/2\right) \mu(dp') + I_0^{\gamma_1/L\epsilon}$$

$$= e^{c_1} \int_{\gamma_1/L\epsilon}^{x_t^\delta} [e^{s_2 p'} - 1] e^{-\alpha_2 p'} \mu(dp') + I_0^{\gamma_1/L\epsilon}, \tag{14}$$

where $c_1 = \alpha_1 \gamma_1 - \gamma_2^2/2 > 0$ for large enough $x_t$, as $\gamma_1$ and $\gamma_2$ are of the same order. Now turning to $p' \in (-x_t^\delta, 0)$, we can use an exact rearrangement of the same argument (noting that $c_1$ can be made arbitrarily small) to get

$$I_{-x_t^\delta}^0 \leq e^{c_1} \int_{\gamma_1/L\epsilon}^{x_t^\delta} [e^{-s_2 p'} - 1] \mu(dp') < 0. \tag{15}$$

Combining (14) and (15) and rearranging as in [30, Theorem 3.2] shows that $I_{-x_t^\delta}^{x_t^\delta}$ is strictly negative in the limit if $s_2 = sL\epsilon$ is chosen small enough, as $I_0^{\gamma_2/L\epsilon}$ can also be made arbitrarily small.

For $I_{-\infty}^{-x_t^\delta}$ it suffices to note that the Gaussian tails of $\mu(\cdot)$ will dominate the exponential growth of $e^{s(\|x^*(p')\|_2 - \|x_t\|_2)}$ meaning the integral can be made arbitrarily small by choosing large enough $x_t$, and the same argument holds for $I_{x_t^\delta}^\infty$. $\qquad\square$

# D  Additional Experimental Details

This section contains additional details for the experiments in Section 6.

## D.1  Stability in High Dimensions

We reproduce the experiment in Figure 2 on an *isotropic* Gaussian in increasing dimension. As length-scales across all principal components are equal, this is a significantly less challenging target to estimate gradients for; though still useful as a benchmark representing very smooth targets. We use a standard Gaussian kernel and the same experimental protocol as for Figure 2. The estimator works slightly better than on the target considered in Figure 2, and performs well up to $d \approx 100$, see Figure 5.

## D.2  Banana target

Following [12, 10], let $X \sim \mathcal{N}(0, \Sigma)$ be a multivariate normal in $d \geq 2$ dimensions, with $\Sigma = \text{diag}(v, 1, \ldots, 1)$, which undergoes the transformation $X \to Y$, where $Y_2 = X_2 + b(X_1^2 - v)$, and $Y_i = X_i$ for $i \neq 2$. We will write $Y \sim \mathcal{B}(b, v)$. It is clear that $\mathbb{E}Y = 0$, and that

$$\mathcal{B}(y; b, v) = \mathcal{N}(y_1; 0, v)\mathcal{N}(y_2; b(y_1^2 - v), 1) \prod_{j=3}^{d} \mathcal{N}(y_j; 0, 1).$$

We choose $d = 8$, $V = 100$ and $b = 0.03$, which corresponds to the 'strongly twisted' 8-dimensional Banana in [12, 10]. The target is challenging due to the nonlinear dependence of the first two dimensions and the highly position dependent scaling within these dimensions.

### D.3 Pseudo-Marginal MCMC for GP Classification

**Model** Closely following [12], we consider a joint distribution of GP-latent variables $\mathbf{f}$, labels $\mathbf{y}$ (with covariate matrix $X$), and hyperparameters $\theta$, given by

$$p(\mathbf{f}, \mathbf{y}, \theta) = p(\theta)p(\mathbf{f}|\theta)p(\mathbf{y}|\mathbf{f}),$$

where $\mathbf{f}|\theta \sim \mathcal{N}(0, \mathcal{K}_\theta)$, with $\mathcal{K}_\theta$ modeling the covariance between latent variables evaluated at the input covariates: $(\mathcal{K}_\theta)_{ij} = \kappa(\mathbf{x}_i, \mathbf{x}'_j|\theta) = \exp\left(-\frac{1}{2}\sum_{d=1}^{D} \frac{(x_{i,d} - x'_{j,d})^2}{\ell_d^2}\right)$ and $\theta_d = \log \ell_d^2$. This covariance parametrisation allows to perform Automatic Relevance Determination. We here restrict our attention to the binary logistic classifier, i.e. the likelihood is given by

$$p(y_i|f_i) = \frac{1}{1 - \exp(-y_i f_i)},$$

where $y_i \in \{-1, 1\}$. Aiming for a fully Bayesian treatment, we wish to estimate the *marginal* posterior of the hyperparameters $\theta$, motivated in [4]. The marginal likelihood $p(\mathbf{y}|\theta)$ is intractable for non-Gaussian likelihoods $p(\mathbf{y}|\mathbf{f})$, but can be replaced with an unbiased estimate

$$\hat{p}(\mathbf{y}|\theta) := \frac{1}{n_{\text{imp}}} \sum_{i=1}^{n_{\text{imp}}} p(\mathbf{y}|\mathbf{f}^{(i)})\frac{p(\mathbf{f}^{(i)}|\theta)}{q(\mathbf{f}^{(i)}|\theta)}, \tag{16}$$

where $\left\{\mathbf{f}^{(i)}\right\}_{i=1}^{n_{\text{imp}}} \sim q(\mathbf{f}|\theta)$ are $n_{imp}$ importance samples. In [4], the importance distribution $q(\mathbf{f}|\theta)$ is chosen as the Laplacian or as the Expectation Propagation (EP) approximation of $p(\mathbf{f}|\mathbf{y}, \theta) \propto p(\mathbf{y}|\mathbf{f})p(\mathbf{f}|\theta)$, leading to state-of-the-art results.

**Experimental details** We here use a Laplace approximation and $n_{\text{imp}} = 100$. We consider classification of window against non-window glass in the UCI Glass dataset, which induces a posterior that has a nonlinear shape [12, Figure 3]. Since the ground truth for the hyperparameter posterior is not available, we initially run multiple hand-tuned standard Metropolis-Hastings chains for 500,000 iterations (with a 100,000 burn-in), keep every 1000-th sample in each of the chains, and combine them. The resulting samples are used as a benchmark, to evaluate the performance all algorithms. We use the MMD between each sampler output and the benchmark sample is computed, using the polynomial kernel $(1 + \langle\theta, \theta'\rangle)^3$. This corresponds to the estimation error of all mixed moments of order up to 3.

**Cross-validation** Kernel parameters are tuned using a black box Bayesian optimisation package[8] and the median heuristic for KMC and KAMH repsectively. The Bayesian optimisation uses standard parameters and is stopped after 15 iterations, where each trial is done via a 5-fold cross-validation of the score matching objective (5). We learn parameters after MCMC 500 iterations, and then re-learn after 2000. We tried re-learning parameters after more iterations, but this did not lead to significant changes. The costs for this are neglectable in the context of PM-MCMC as estimating the marginal likelihood takes significantly more time than generating the KMC proposal.

### D.4 ABC MCMC

In this section, we give a brief background on Approximate Bayesian Computation, and how KMC can be used within the framework. We then give details of the competing approach in the final experiment in Section 6, including experimental details and an analytic counterexample.

**Likelihood-free Models** Approximate Bayesian Computation is a method for inference in the scenario where conditional on some parameter of interest $\theta$, we can easily simulate data $x \sim f(\cdot|\theta)$, but for which the likelihood function $f$ is unavailable [6]. We however have data $y$ which assume to be from the model, and we have a prior $\pi_0(\theta)$. A simple ABC algorithm is to sample $\theta_i \sim \pi_0(\cdot)$ (or any other suitable distribution), simulate data $x_i \sim f(\cdot|\theta_i)$, and 'accept' $x_i$ as a sample from the approximate posterior $\pi_\epsilon(\theta|y)$ if $d(y,x) \leq \epsilon$. This procedure can be formalised by defining the approximate likelihood as

$$f_\epsilon(y|\theta) \propto \int g_\epsilon(y|x,\theta)f(x|\theta)\mathrm{d}x, \tag{17}$$

where $g_\epsilon(y|x,\theta)$ is an appropriate kernel that gives more importance to points for which $d(y,x)$ is smaller. In the simple case above $g_\epsilon(y|x,\theta) = \mathbf{1}_{\{d(y,x)\leq\epsilon\}}$. The ABC posterior is then found using $\pi_\epsilon(\theta|y) \propto f_\epsilon(y|\theta)\pi_0(\theta)$. Often $g_\epsilon$ is based on some low-dimensional summary statistics, which can have both advantages and disadvantages.

**Likelihood-free MCMC** There are many different way to do ABC, and clearly not all involve Markov chain Monte Carlo. If the posterior however is not similar to the prior, and if $\theta$ is more than three or four dimensional, MCMC is a sensible option. Since the likelihood (17) is intractable, typically algorithms are considered for which an approximation to either the likelihood, or the ABC posterior are used either in constructing proposals, defining Metropolis-Hastings acceptance rates, or both. We focus here on samplers which target $\pi_\epsilon(\theta|y)$ directly, c.f. [5].

**Pseudo-Marginal Metropolis-Hastings** Similar to the approach taken in Section D.3, we here accept proposals $\theta' \sim Q(\theta,\cdot)$ where $Q$ is some proposal mechanism (i.e. KMC), via replacing the likelihood with an unbiased estimate. We accept according to the ratio

$$\tilde{\alpha}(\theta,\theta') = \frac{\tilde{\pi}_\epsilon(\theta'|y)Q(\theta|\theta')}{\tilde{\pi}_\epsilon(\theta|y)Q(\theta'|\theta)}, \tag{18}$$

where $\tilde{\pi}_\epsilon(\theta|y) = \pi_0(\theta)\tilde{g}_\epsilon(y|\theta)$, and

$$\tilde{g}_\epsilon(y|\theta) = \frac{1}{n_{\mathrm{lik}}} \sum_i g_\epsilon(y|x_i,\theta), \ \ \{x_i\}_{i=1}^{n_{\mathrm{lik}}} \sim f(\cdot|\theta_i)$$

is a simple Monte Carlo estimator for the intractable likelihood (17). Since it is easy to simulate from $f$ then $\tilde{g}_\epsilon(y|\theta)$ is typically easy to compute. As with other general Pseudo-Marginal schemes, and as mentioned below the KMC acceptance (3), it is crucial that if $\theta'$ is accepted, the same estimate for $\tilde{\pi}(\theta'|y)$ is used on the denominator of the Hastings ratio in future iterations until the next proposal is accepted for the scheme, c.f. [3, Table 1].

We can directly adapt KMC to the ABC case via plugging in the estimated likelihood $\tilde{g}_\epsilon$ in the KMC acceptance ratio (3).

**Synthetic Likelihood Metropolis-Hastings** Following [26], one idea to approximate the intractable likelihood is to draw $n_{\mathrm{lik}}$ samples $x_i \sim f(\cdot|\theta_i)$, and fit a Gaussian approximation to $f$, producing estimates $\hat{\mu}$ and $\hat{\Sigma}$ for the mean and covariance using $\{x_i\}_{i=1}^{n_{\mathrm{lik}}}$. If the error functon $g_\epsilon$ is also chosen to be a Gaussian (with mean $y$ and variance $\epsilon$), then the marginal likelihood $f_\epsilon(y|\theta)$ can be approximated as

$$y|\theta \sim \mathcal{N}\left(\hat{\mu}, \hat{\Sigma} + \epsilon^2 I\right)$$

The likelihood is essentially approximated by a Gaussian $f_G$, producing a synthetic posterior $\pi_s(\cdot)$, which is then used in the accept-reject step. Clearly some approximation error is introduced by the Gaussian likelihood approximation step, but as shown in [26], it can be a reasonable choice for some models.

**Hamiltonian ABC** Introduced in [8], the synthetic likelihood formulation is used to construct a proposal, with the accept-reject step removed altogether. Hamiltonian dynamics use the gradient $\nabla \log \pi(\theta)$ to suggest candidate values for the next state of a Markov chain which are far from the current point, thus increasing the chances that the chain mixes quickly. Here the gradient of the log-likelihood is unavailable, so is approximated with that of a Gaussian (since the map $\theta \to (\mu, \Sigma)$

is not always clear this is done numerically, using a stochastic finite differences estimate of the gradient, SPAS [8, Sec. 4.3, 4.4]), giving

$$\nabla \log \pi(\theta) \approx \sum_{i=1}^{n_{\text{lik}}} \nabla \log f_G(y_i|\hat{\mu}, \hat{\Sigma}) + \nabla \log \pi_0(\theta).$$

Since there is no accept-reject step, the synthetic posterior is also the target of this scheme (although there is also further bias introduced by discretisation error), but the introduction of gradient-based dynamics is hoped to improve mixing and hence efficiency of inferences compared to random-walk type schemes.

**A Counter-example**   We give a very simple toy model to highlight the bias introduced by the Hamiltonian ABC sampler. Consider posterior inference for the mean parameter in a log-Normal model. Specfically, the true model is

$$\mu \sim \mathcal{N}(\mu_0, \tau_0),$$
$$y|\mu, \tau \sim \log \mathcal{N}(\mu, \tau),$$

where the precision $\tau$ and hyperparameters $\mu_0, \tau_0$ are known. The model is in fact conjugate, giving a Gaussian posterior

$$\mu|y \sim \mathcal{N}\left(\frac{\tau_0 \mu_0 + \tau \sum_i \log x_i}{\tau_0 + n\tau}, \tau_0 + n\tau\right).$$

If we introduce a Gaussian approximation to the likelihood, then the mean and precision of this approximation $f_G$ are (empirical estimates for)

$$\mu_G = e^{\mu + 1/2\tau}, \ \ \tau_G = 1/\text{Var}[Y_i] = \frac{e^{-2\mu - 1/\tau}}{e^{1/\tau} - 1},$$

which depend on the current value for $\mu$ in the chain. The resulting synthetic posterior is no longer tractable, but since it is one dimensional we can approximate it numerically. Using $\mu_0 = 0$, $\tau_0 = 1/100$, $\epsilon = 0.1$ and $\tau = 1$ then the true and approximate posteriors for 100 data points generated using the truth $\mu = 2$ are shown in Figure 6 (left). This is a proof of concept that a likelihood with a positive skew being approximated by a Gaussian introduces an upwards bias to the posterior.

**Experimental details**   The simulation study in Section 6 uses a slightly more complex and multi-dimensional simulation example: a 10-dimensional multivariate skew-Normal distribution, given by

$$p(y|\theta) = 2\mathcal{N}(\theta, I) \Phi(\langle \alpha, y \rangle)$$

with $\theta = \alpha = \mathbf{1} \cdot 10$. In each iteration of KMC, the likelihood is estimated via simulting $n_{\text{lik}} = 10$ samples from the above likelihood. We use the mean of all samples as summary statistic, and a Gaussian similarity kernel $g_\epsilon(y|x, \theta)$ with a fixed $\epsilon = 0.55$. Both KMC and HABC use a standard Gaussian momentum, a uniformly random stepsize in $[0.01, 0.1]$ and $L = 50$ leapfrog steps. HABC is used with the suggested 'sticky random numbers' [8, Section 4.4], i.e. we use the same seed for all simulations along a single porposal trajectory. Both algorithms are run for $200 + 5000$ MCMC iterations. KMC then attempts to re-learn smoothness parameters, and stops adaptation. Burn-in samples are discarded when quantifying performance of all algorithms.

**Friction, mixing, and number of simulations**   HABC is used in its 'stochastic gradient' [7] and has a 'friction' parameter that we estimate using a running average of the global covariance of all SPAS gradient evaluations, [8, Equation 21]. Note that we ran HABC with both the friction term included and removed, where we found that adding friction has severely negative impact on mixing, where not adding friction results in a wider posterior (with the same bias). Figure 4 (middle, right) show the results *without* friction, Figure 6 shows the same plots with friction. We refer to our implementation for further details.

Due to the gradient estimation in every of the $L = 50$ leap-frog steps, e*very* MCMC proposal for HABC requires $2L = 100$ simulations to be generated. In contrast, KMC only requires a single simulation, for evaluating the accept/reject probability (3). We leave studying the exact trade-offs of KMC's learning phase and its ability to mix well as compared to HABC to future work.

Figure 6: **Left:** Counter example showing posterior and its synthetic approximation for a simple toy model **Middle/right:** The same results as in Figure 4, but here we also show performance of HABC with added friction, which has a severely negative impact on mixing.

## Footnotes

[5]In this paper, we solely use the Gaussian kernel, whose spectrum decays exponentially fast.

[6] $C$ is the empirical covariance of the feature derivatives $\dot{\phi}_{x_i}^\ell$.

[7]We use the open-source implementation provided at `https://github.com/jcrudy/choldate`

[8]We use the open-source package `pybo`, available under `https://github.com/mwhoffman/pybo`