[Reviews · NeurIPS 2015]

Submitted by Assigned_Reviewer_1

The quality is very good, I think the development of the theory and the proposed efficient solvers to overcome the computational burden of the algorithm. It is written clearly, making me read smoothly from beginning through the end. Also, though related to some existing algorithms such as the Hamiltionan ABC, I believe the originality of paper is high, and it would potentially bring significant impacts in HMC literatures where the gradient is not available.

One concern I have is that this paper focus on the setting where the gradient is unavailable, though mentioned in the paper, I wonder in the setting of large data where the gradient is computationally expensive to compute, would the algorithm still feasible? It seems we still need the whole data to calculate the acceptance probability, making it expensive to use in this setting.

Another comments relates to the comparison of the proposed KMC with the original HMC, where the gradient is available. It seems from the experiments that HMC is better than KMC in this setting. I wonder why this happens? Would the KMC be more general than HMC because it is defined in the RKHS? Any theory to explain the reason?

Another comparison is about the KMC lite and KMC finite. I'd like to see some comparison between them, any conclusions on which one is better?

After feedback: I think the idea in this paper is new, and I like it. I believe this is a good paper, and encourage the authors revise the paper according to the reviews.
Summary: This paper proposes a Hamiltonian MCMC without the knowledge of gradient information in the computation in the RKHS setting. The paper is well written, and the theory is well established, filling an important gap in the HMC literature.

Submitted by Assigned_Reviewer_2

This paper proposed an interesting method which uses kernel density estimation as a Hamiltonian surrogate, in order to improve ESS and acceptance rate of random walk based method when gradient is not available. The algorithm makes sense to me and the experimental results confirms the intuition and shows better performance compared with RW, KAMH. Furthermore, the performance gets closer to HMC as more samples are seen.

One thing that I was a bit worried is that how the surrogates handle the area which were not explored before. This may bring serious issues such as the break of ergodicity. However, it appears that KMC lite naturally reduces to a random walk in these areas, and therefore ergodicity is maintained. This is very nice.

Although the experiemnt does not involve very large dataset, the analysis of the given datasets are comprehensive and convincing.

There are some minor typos but overall the writing is good and well organized.

Typos:

1. line 103: in a RKHS --> in an RKHS 2. line 150: <> --> missing a sub H 3. line 185-189: H_k, \hat{H} should be \hat{H}_k? 4. line 242: H^m --> R^m.
Summary: An interesting method which uses kernel density estimation as a Hamiltonian surrogate, in order to improve ESS and acceptance rate of random walk based method when gradient is not available.

Submitted by Assigned_Reviewer_3

The authors present a way to perform hamiltonian monte carlo for target distributions with difficult to compute gradient functions.

They use a kernel-based approximation to the log-likelihood function that admits a tractable gradient.

This procedure admits an HMC-like proposal, and they use a metropolis correction to ensure the chain leaves the correct distribution invariant.

How would you compare ESS/likelihood evaluation between KMC and standard HMC?

I know it's a bit apples and oranges, but I'm trying to get an idea of how expensive the KMC procedure is.
Summary: This is a 'light' review: the paper is clear, appears to be well executed, and, as far as I know, a novel contribution to MCMC methodology.

Submitted by Assigned_Reviewer_4

Summary: The paper is primarily concerned with a Bayesian inference setting in which only unbiased estimates of an unnormalized density are available. The authors propose using kernel methods to approximate the unnormalized density as well as the the gradient of the log-probability in order to perform approximate HMC. The paper uses approximate versions of score matching to approximate the log-probability. Because standard score matching scales poorly with the number of samples being used (that is, the number of samples in the Markov chain thus far), the authors propose only using a random subset of the the samples (KMC lite) or a finite approximation of the infinite-dimensional RKHS (KMC finite).

The paper has a number of intriguing ideas and potentially very useful methodologies for pseudo-marginal and ABC-type settings. However, there a substantial issues in the presentation of the methods and in the experimental validations.

Major issues: 1. A crucial idea in the paper is to use (approximate) score matching to construct a nonparametric estimate to the (gradient of the) log-probability. However, the authors only describe score matching in the appendix, but at the same time repeatedly refer to the key score matching equation in that appendix. This is rather poor form, and room for the content of appendix A.1 needs to be found in the main paper. More generally, the authors have a bad habit of leaning heavily on the reader to understand a number of other recent papers, in particular their refs [12, 13, 14], and to a lesser extent [3]. I'm sympathetic to the fact that the NIPS paper format offers very limited space, but some of the relevant material (e.g., the details of the experimental setup of [12]), could easily to put in the supplementary material. Extra room could be found by, e.g., removing the proof sketch of Prop 3 and the useless paper outline at the end of the intro.

2. The authors state (line 94+): "we generally assume that [the density of interest] \pi is intractable, i.e. that we can neither evaluate \pi(x) nor \grad \pi(x) for any x, but can compute unbiased estimates of \pi(x)." However, when presenting their KMC method, eq. (3) makes use of the true Hamiltonian, i.e., it requires access to \pi(x). In the experiments section they state they are in fact using PM-MCMC in place of (3), which is fine given the working assumptions. However, this is never made clear earlier, and, moreover, the authors never bother to state the PM-MCMC acceptance criterion they are using.

3. The experiments were difficult to follow and, insofar as I could follow them, were unconvincing.

a. Stability in high dims: it was unclear how many samples were used (i.e., the value of t), and how these samples were generated since apparently there was "no MCMC yet." Targeting a standard Gaussian here is extremely uninteresting and not even close to representative of what a real high-dimensional target might look like. Placing a distribution over the eigenvalues of the Gaussian covariance, then generating a number of Gaussians from that distribution would be much more reasonable. Also, it's odd to start from all the origin since in high dimensional centered Gaussians the origin is in a region of increasingly low probability.

b. Mixing on a synthetic example: why only investigate mixing? I would expect burn-in to be more problematic since early samples used to estimate pi(x) are going to be unrepresentative. Thus, understanding how KMC performs during burn-in is crucial, and the authors themselves emphasize that the lite and finite versions will perform very differently while in the tails. Small point: min/median/max ESS would be more informative than mean ESS.

c. GP classification: This task seems ok, though it would be good to include further experimental details (e.g., model used, source of benchmark samples) in the supplemental material. How much additional computation is required for the selection of kernel width via x-validation? Needing to run this x-validation seems potentially problematic. Why was 45% acceptance targeted? Why does the KMC MMD stop decreasing after ~3000 iterations in Fig 4(left)? Was KMC finite also tested for this problem?

d. ABC: I found it extremely difficult to follow the set-up here. What is the "KMC-ABC" algorithm being tested? (As in 3c, Alg 1 as written is not applicable.) I eventually figured out that data was sampled from p(y | alpha = theta = 10 * \vec 1), then theta was inferred using the two ABC algorithms. But this was not clear on first reading and further details are missing. E.g., how many data points were used to form the posterior? Even more importantly, the skew-normal distribution used to test KMC-ABC is very simple and unrepresentative of the problems ABC is applied to.

Minor points: 1. The fact that KMC lite falls back to a random walk seems like a really nice feature of that approach and the sort of important detail that might convince a practitioner to try out KMC ("when the method doesn't know what to do, it falls back to method I understand will do something sensible!"). Therefore, it would be good to add a sentence at line 301 explaining how/why this reversion to RW MH occurs.

2. It is of course well-known that choosing reasonable kernel hyperparameters is very important, since they typically encode scale/smoothness assumptions. Particularly in the very complicated models for which only unbiased estimate of pi are available or for which only forward simulations can be performed, the smoothness may vary substantially throughout the parameter space. The paper only addresses these sorts of issues in passing at the end of section 5.

Other comments: Kamiltonian Monte Carlo is a terrible name. Why not just "kernel HMC"?

l.126: missing comma in Hamiltonian l.224: with => are given by

RESPONSE TO AUTHORS: I appreciate the authors' thoughtful response and efforts to redo the Gaussian experiments. However, there are substantial enough issues with the presentation of the methods and the experiments that I think the paper should be re-reviewed. I very much like the paper - the ideas in the paper are very interesting and they deserve to be presented in a clear manner so that the community can benefit from them as much as possible.
Summary: While the paper is original and its methods may prove to be quite useful, in its current form, there is a lack of clarity and compelling experimental validation.

Submitted by Assigned_Reviewer_5

This paper proposes an extension of kernel adaptive MCMC using Hybrid Monte Carlo (HMC).

The proposed idea is quite appealing - drawing samples from the posterior over model parameters with efficiency close to HMC without knowledge of the gradient of the log-density, or the log-density itself provided that it can be estimated unbiasedly. This work is particularly important in modern day applications where more and more we encounter models with intractable likelihoods, so I think that this work deals with an important class of problems.

The paper builds on previous work where the exploration was characterized by a random walk in the RKHS. I believe this new paper proposes ideas that are novel enough to deserve being accepted at NIPS.

Here are some more comments on the paper:

- maybe this is trivial, but I think it's worth mentioning that the leap-frog integrator for the proposed Hamiltonian is symplectic

- some convergence analysis (e.g., Potential Scale Reduction Factor) would have been useful to quantify the rate of convergence of the proposed samplers

- the idea of using vanishing adaptation is excellent. I completely understand the limitation of space, but it would have been nice to see more about this in the experiments. In other words it would have been interesting to see how "automatic" and efficient the proposed KMC can be
Summary: This work extends previous work on kernel adaptive MCMC; I think that the degree of novelty compared to previous work is enough.

The paper is well presented and realized, and I believe it should be considered for publication in this venue.

Author Feedback
Author rebuttal: We thank the reviewers for their constructive feedback.

Reviewer 2.
Experiment critique:
3a. We point out that Figure 2 clearly states the number of samples/basis functions m=n with the numbers on the axis labels. We agree that the Gaussian target is somewhat simple, however, the point here was to exactly have a very simple (Gaussian-smooth) target to quantify estimation quality in higher dimensions in a controlled environment, and leave more realistic targets to the other experiments on real data. Following the reviewer's excellent suggestion, we repeated the experiment: We placed a Gamma distribution on the eigenvalues of the target covariance, rotated with a random orthogonal matrix, and quantified stability in increasing dimensions. As this is a significantly more challenging target, the results naturally degrade when a Gaussian kernel with a single length-scale is used. Choosing an appropriate kernel function however somewhat restores the reported performance: a rational quadratic kernel (infinite Gamma-mixture of Gaussians, straightforward random features) achieves (e.g. for d=16 as in Fig 2) trajectory agreements of 0.2, 0.4, 0.7 for n=2000, 5000, 10000 respectively, with slightly larger error bars than in Fig 2. This allows for similar conclusions as in the paper (KMC scales up to medium on Laptops), with the addition of the intriguing point that KMC works on targets with 'non-singular' smoothness. In the final version, we will replace the simple Gaussian example with this new experiment. Note: the statement that we initialise all trajectories on the origin was an old statement slipping through, the original experiment was done starting at a random sample.
3b. X-Validation computation is the same as a few hundred MCMC iterations -- not significant for the dimensions we considered. In practice we use schedules of "hot-started" global black-box optimizers that allow to update hyper-parameters with just a few iterations.
General: The presented experiments were carefully selected to make clear the key properties of the proposed method: stability on smooth and constrained (Gaussian/Banana) targets in up to medium dimensions, improved mixing in (real-world) situations where the state-of-the-art is random walk sampling, and no-bias compared to other published ABC methods (indeed skew-normal is simple, but it was used only to demonstrate no-bias: even for this simple example competing methods suffer bias). We agree with the reviewer that it will be exciting to explore performance in additional settings, however our present experiments illustrate all major performance claims within the constraints of the NIPS page limit.
Location dependent curvature in the parameter-space is an extremely hard problem (for both KMC and plain HMC!) and intentionally left for future work.
We will consider name change to K-HMC.
We are slightly surprised by the score given the review text: A large part of the structural critique (2 of 3 points) are addressed with relatively few changes to the main text, following the reviewer's helpful suggestions (score matching & intractable targets).
We hope to have convinced reviewer 2 to revise his score.

Reviewer 3.
We agree that the idea of surrogate energy functions has appeared earlier, and cite the method that is closest to ours, [16]. As mentioned in the intro, the GP surrogate requires evaluations of the *true* log-pdf of the target, whereas KMC learns the target from samples only, and allows for efficient updating. In addition, an RKHS-based infinite-dimensional exponential family model is used for modelling targets (such a model has been demonstrated to have good performance in high dimensions). Our main contributions are novel, to the best of our knowledge.

Reviewer 4
The large-scale data setting is an interesting application for KMC. As evaluation of the surrogate gradient is independent of the dataset, we expect speed-ups, order O(leapfrog_steps*N), when applied to large N. The costs however remain linear in N due to the accept reject ratio. Combinations of KMC with approximate accept-reject methods are a topic for future work.

Reviewer 5 & reviewer 1
We agree that quantifying vanishing adaptation is a nice experiment, and indeed we have run such experiments: they show that KMC converges (first 3 moments) faster during burn-in, when compared to adaptive MH with the same learning rate. This is as expected: as the energy function is learned, better mixing is obtained. Due to space constraints, and as the results are very similar to the plots presented, they were not in the paper. We will add a plot in the Appendix.
We also ran experiments where we estimated gradients of a bimodal target using samples from one mode only, and then stopped adaptation. As KMC lite degrades to a random walk, it is able to explore the previously "unseen" mode despite the fact that the gradient estimate is flat on the second mode. We will add these illustrations to the Appendix.